# Laboratory Investigation and Numerical Modelling of Concrete Reinforced with Recycled Steel Fibers

**DOI:** 10.3390/ma14081828

**Published:** 2021-04-07

**Authors:** Małgorzata Pająk, Małgorzata Krystek, Mateusz Zakrzewski, Jacek Domski

**Affiliations:** 1Department of Structural Engineering, Silesian University of Technology, Akademicka 5, 44-100 Gliwice, Poland; malgorzata.krystek@polsl.pl; 2Faculty of Civil Engineering, Environmental and Geodetic Sciences, Koszalin University of Technology, Sniadeckich 2, 75-453 Koszalin, Poland; mateusz.zakrzewski@tu.koszalin.pl (M.Z.); jacek.domski@tu.koszalin.pl (J.D.)

**Keywords:** concrete, recycled steel fibers, flexural tests, numerical modelling

## Abstract

In the last decades, fiber reinforced concrete have emerged as the possible key to revolutionize civil engineering. Among different types of fibers employed in concrete technology to date, the application of recycled steel fibers produced from end-of-life car tires appears to be a viable approach towards environmentally friendly construction. In this study, we demonstrate the laboratory research and numerical analysis of concrete reinforced with waste steel fibers recovered during the recycling process of end-of-life car tires. Concrete mixes with the following fiber contents: 0.5%, 0.75%, 1.0%, 1.25%, and 1.5% per volume were prepared and then tested in three-point bending conditions. The laboratory investigation revealed highly boosted properties of concrete under flexure. We further performed the finite element method (FEM) analysis of 2D models using Atena software in order to develop a material model allowing the numerical modelling of recycled steel fibers reinforced concrete (RSFRC) behavior. The parameters of RSFRC material model have been modified using the inverse analysis until matching the experimental performance of the material. The results, being in good agreement with the laboratory investigation, have indicated a high potential of RSFRC for real scale construction applications.

## 1. Introduction

The search for technological solutions to the ever-increasing demand in concrete production represents one of the greatest challenges the concrete industry is currently facing. During the last decades, a great effort has been devoted to the application of waste materials as concrete components in the field of construction. The main advantages of this strategy lie with the potential environmentally friendly applications by reusing waste and reducing the energy consumption associated with the production of new materials. In particular, a prime example is represented by the replacement of concrete aggregate with ceramic waste [1] or concrete rubble [2].

In order to enhance the post-cracking residual tensile strength and ductility, the randomly distributed fibers are added to concrete to create a composite material [3,4]. Fiber reinforced concrete (FRC) is characterized by improved fatigue resistance [5], toughness [6], durability [7], and thermal and fire resistance [8] with comparison to concrete. Fibers are also effective in improving the mechanical parameters of concrete subjected to high strain rates associated with dynamic loads [9]. The variety of short, randomly distributed fibers was developed to be applied as concrete reinforcement and released on the market in the last decades [6,10,11].

Interestingly, waste materials may be capable of replacing fibers in fiber reinforced concrete (FRC) or even outperform them. Within the recycling process, the tires are subjected to the mechanical treatment and rubber granulate, as well as steel cord, also called recycled steel fibers (RSF), are produced. Notably, steel cord differs significantly from commercially produced fibers in terms of the length, diameter, and irregular shape [12]. Indeed, since different types of car tires can be possessed and recycled, the length and diameter of the steel cord may vary considerably, thus making RSF a hybrid fiber mix. Such hybrid fiber mixes proved to be more effective in concrete than just one particular type of fiber [13]. Moreover, the selected properties of concrete may be greatly benefited by the irregular shape of RSF as well as by their capability to match the shape of aggregate grains.

The successful applications of the recycled steel fibers in concrete have been already proved by a number of research groups [14,15,16,17,18,19]. Indeed, it has been shown that the RSF may become an extremely promising alternative for industrial fibers; however, most likely, the higher RSF loadings should be incorporated into concrete to achieve the comparable effects in terms of mechanical properties of concrete [17]. Further comparison has revealed that RSF also match the shear performance of industrial fibers [15]. Nevertheless, the lack of recommendations and designing protocols for RSF concrete limits its real applications to a large extent. Therefore, to bridge the gap between the laboratory-scale tests and commercial applications, further research developing a mathematical model is essential.

Here, we present a novel material model allowing the numerical simulation of RSFRC structures. First, we performed the laboratory investigation on the effect of the recycled steel fibers addition on flexural and compressive parameters of concrete. The results served as the base for the material model applied in further numerical simulation. The numerical analysis performed using the linear material model of FRC from Model Code 2020 [20] revealed the need for further calibration of the material model. The numerical simulation of tested beams was performed using Atena Studio v5 software [21], which is an extremely powerful tool for non-linear analysis of FRC. The numerical analysis was performed on 2D models. In the Atena Studio v5 software we employed the inverse analysis to meet the stress-strain relationship from experimental study.

In the literature there can be found papers dealing with a numerical approach based on experiments of different kinds of concrete reinforced with fibers [6,22,23]. The numerical simulations of FRC are based mainly on sophisticated numerical tools. The aim of the present paper was to prepare a useful and relatively uncomplicated numerical tool for performing the analysis of concrete reinforced with RSF. The Atena Studio v5 appeared to be the best program for this purpose. By modification of the material model based on laboratory investigation, the RSFRC structures can be easily analyzed. Thus, just knowing the residual flexural tensile strengths from the laboratory investigation the RSFRC structures can be analyzed. Notably, this approach may open up new avenues for the technological applications of RSFRC in construction.

## 2. Materials and Methods

### 2.1. Materials

To produce recycled steel fiber reinforced concrete (RSFRC), waste sand (fractions ranging from 0.125 to 4 mm) from a local (Pomerania, Poland) mine of natural aggregates was used. The successful application of the waste sand in concrete production has been already proved within the studies performed at the Koszalin University of Technology [24]. Fine aggregate and coarse aggregate were added according to standard EN 1766 [25] using curve “A” (Figure 1). As a binder, Portland cement CEM I 42.5 R was employed in an amount 350 kg/m^3^ with w/c ratio 0.55. The water used to produce concrete came from the drinkable water supply; hence, in accordance with EN 1008, there was no need for further chemical characterization of water composition [26]. The exact composition of the RSFRC is presented in Table 1.

The RSF was used as fiber reinforcement (Figure 2). As previously mentioned, the length and diameter of the steel cord may differ considerably due to the wide variety of recycled tires; therefore, in order to determine the basic parameters of the cord, one batch of 106 fibers randomly taken from a bag supplied by the manufacturer was analyzed. As revealed by the measurements, the fibers with a diameter between 0.20 and 0.35 mm were a main product accounting for 90% of the batch (Table 2). The average tensile strength of steel cord was 1788.5 MPa; however, it should be pointed out that our attempts to characterize the tensile strength of the fibers shorter than 30 mm failed due to sliding out of the jaws of the testing machine. The slenderness of the steel cord is displayed in Figure 3, with the normal distribution function (red curve).

### 2.2. Experimental Procedure 

The fresh concrete (RSFRC) was tested in accordance with EN 12350-4 standard [27]. Concrete was placed in a rectangular container with the square base of 200 × 200 mm^2^ and the height of 400 mm. The container was then placed on a vibrating table. After vibration, the change in yield was measured. The degree of compactability c was determined as follows:c = h/(h − s)(1)
where h is the internal height of the container, mm, and s is the mean value, in mm, of the distance between the surface of the compacted concrete and the upper edge of the container measured along each container wall.

The residual flexural tensile strength tests were carried according to EN-14651 [28], on beams with dimensions of 150 × 150 × 600 mm^3^. The samples were subjected to the three-point bending test. Prior to testing, the samples were incised in the central part to a depth of 25 mm (Figure 4). The load was applied directly above the incision in a continuous manner with a variable displacement rate. In the first phase, to the width of CMOD = 0.1 mm, the load was applied at a displacement rate of 0.5 mm/min. After exceeding this value, the displacement rate was decreased to 0.2 mm/min. The test was conducted until the deflection of 4 mm was reported. The detailed description of the procedure for determining the residual flexural tensile strength is revealed in Section 3.1. 

The deflection and crack width were recorded using the SAD 256 data acquisition system with the set consisting of a few electrofusion sensors. The sensor with a range of 20 mm, rigidly mounted in the machine frame, measured the deflection in the middle of the beam span, while the sensor with a range of 10 mm was used to measure the crack width (CMOD). The sensors were mounted in custom-made frames (Figure 4).

Fifteen samples were manufactured and examined, i.e., three samples for each fiber volume ratio: 0.5%, 0.75%, 1.0%, 1.25%, and 1.5%.

The compressive strength tests were conducted on six cubed specimens with dimensions 150 × 150 × 150 mm^3^. The load was applied with a constant loading rate of 4 kN/s. 

### 2.3. Experimental Results

The results of the degree of compactability experiments are shown in Table 3. According to EN-206 [29], the mixtures 0.5%, 0.75%, and 1.0% can be classified as fresh concrete consistency class C2, while the mixtures 1.25% and 1.50% can be assigned to class C3.

The average load-CMOD curves recorded during the flexural tests are reflected in Figure 5. It is clearly visible that the fibers improved the post-crack response of FRC to flexural loading. Notably, the effectiveness of the fibers increases with the increase of the amount of fibers.

Surprisingly, the results obtained within this study differ significantly from the results reported in the previous work [18], where the increase of the volume ratio of RS led to the considerable spread of the results. Here, even for the highest RSF loading, i.e., 1.5%, the spread of the results may be considered as marginal (see Figure 6). This fact may be attributed to the size and type of the aggregate and the slump flow of the mix in case of the concrete described in the work [18]. The size of the coarse aggregate (16 mm) and the slump flow was bigger than in the case of the results presented herein. Generally, the proceeding of the load-CMOD curves noted in the case of those two laboratory investigations were comparable for the related fiber volume ratios, even though the concrete with the higher compressive strength (around 50 MPa) was tested in [18] (see Figure 7).

The shape of the load-CMOD curves remains relatively unaltered for concrete mixes 0.5%, 0.75, and 1.0%, with the f_R.3_/f_R.1_ ratio of ca. 0.6 (between 0.51 and 0.66) corresponding to letter “a” in MC2010 classification [20]. Notably, the f_R.1_/f_LOP_ coefficients for these RSF loadings are comparable and amount to ca. 0.7. A highly noticeable shift of the coefficients value occurred for the mixes 1.25% and 1.5%, with the f_R.3_/f_R.1_ coefficient of approximately 0.8 (letter “b” according to MC2010 [20]) and the f_R.1_/f_LOP_ ratio of 0.89 and 1.06, respectively. Roughly speaking, RSFRC was classified in class 2a for mixes 0.5% and 0.75% and class 3a, 3b, and 5b for mixes 1.0%, 1.25%, and 1.5%, respectively. Moreover, the ratios were f_R.1_/f_LOP_ > 0.4 and f_R.1_/f_LOP_ > 0.5, indicating that the fiber reinforcement used within this study can be used as the replacement for conventional reinforcement. The detailed results from the laboratory investigation are presented in Table 4. 

The effect of fiber addition on the compressive strength of concrete can be considered as marginal and these findings are in line with the previous observations of various researchers [14,15,16,17]. The average compressive strength was determined as ca. 40 MPa for all concrete mixes. Since concrete is a heterogeneous material, the variations of the measured values should not be recognized as the effect of steel fibers. 

## 3. Numerical Modelling

### 3.1. Material Model

In general, concrete features with low tensile strength and quasi-brittle behavior lead to high vulnerability to cracking. The application of fiber reinforcement may enhance the mechanical parameters of concrete in tension; however, the material model considering the post-cracking stage in tension needs to be implemented.

The applied material model considers the influence of the fibers on the stress-strain relationship of concrete in tension. This relationship is determined on the basis of the three-point flexural tensile tests performed according to EN-14651 [28]. As stated in the recommendation [28], the residual flexural tensile strengths have to be determined from the load versus CMOD diagram (Figure 8). The residual flexural tensile strengths are specified for CMOD1 = 0.5 mm (f_R.1_), CMOD2 = 1.5 mm (f_R.2_), CMOD3 = 2.5 mm (f_R.3_), and CMOD4 = 3.5 mm (f_R.4_). The further crucial parameter is the flexural tensile strength at the limit of proportionality (f_L_), which is defined as the highest value in the interval of 0.05 mm. The residual flexural tensile strengths and the limit of proportionality can be calculated as follows:f_R.i_ = (3 × F_R.i_ × l)/(2 × b × hsp^2^); f_L_ = (3 × F_L_ × l)/(2 × b × hsp^2^)(2)
where h_sp_—distance between the notch tip and the top of the specimen; F_R.i_—load corresponding to CMODi; F_L_—load corresponding to the limit of proportionality; l—span of the beam; and b—width of the specimen. 

The load value f_L_ is determined by drawing at a distance of 0.05 mm a line parallel to the load axis of the load-CMOD or load-deflection diagram. The highest load value in the interval of 0.05 is then considered as f_L_. 

Considering the parameters obtained from laboratory investigation, the linear material model for FRC (Figure 9), widely discussed in [3], was proposed in the Model Code 2010 [20]. The reference values of this material model are defined using the residual flexural tensile strengths according to the following equations: f_Fts_ = 0.45 × f_R.1_(3)
f_Ftu_ = f_Fts_ − w_u_/CMOD_3_ (f_Fts_ − 0.5 × f_R.3_ + 0.2 × f_R.1_) ≥ 0(4)
where w_u_—maximum crack opening accepted in structural design.

The maximum crack opening was assumed here as 2.5 mm. The l_cs_, which is the structural characteristic length corresponding to w_u_, was defined as 250 mm. Any other data required for the evaluation of the model parameters were calculated according to MC2010 [20]. The modulus of elasticity and tensile strengths were defined as follows:E_c_ = 21.5 × (0.1 × f_cm_)^1/3^(5)
f_ctm_ = 0.3 (f_cm_)^2/3^(6)

The parameters of the basic material model (Figure 2) with the data from the laboratory investigation crucial for their calculation are summarized in Table 5.

The numerical simulations of the notched beams performed with the basic material model revealed the need to calibrate the model parameters.

Based on the inverse analysis the parameters of the final material model were determined and schematically presented in Figure 10. The inverse analysis of the notched beams fabricated using concrete reinforced with the following volume ratios of the RSF: 0.5%, 0.75%, 1%, 1.25%, and 1.5% was performed. For all mixes, the maximum tensile stress that can be achieved in the FRC (σ_1_) had to be dependent on the limit of proportionality f_L_ instead of the tensile strength f_ctm_. In that case, the compliance of the most precise results was obtained. This phenomenon can be attributed to the fact that the limit of proportionality was possessed directly from the flexural tests. The 45.5% of the limit of proportionality was appropriate for all mixes. In the final model the maximum tensile stress was assigned according to the equation
σ_1_ = 0.455 × f_L_(7)

The stress at breakpoint of the linear material model (σ_2_) was dependent on the residual flexural tensile strength determined for the CMOD equal to 2.5 mm (f_R.3_). The value had to be limited to the 38% of its value and was derived from the equation
σ_2_ = 0.38 × f_R.3_(8)

The strain ε_1_ corresponding to the stress σ_1_ was assigned with the assumption of the elastic behavior of FRC. In all fibers amounts, the strain ε_2_ corresponding to the breaking point of the linear stress-strain relationship was estimated as 5%. The stress achieved 0 value for the strain equal to 160%. The final value of the parameters of the material model with the data from the laboratory investigation needed for their calculation are listed in Table 6.

The schematic comparison of the basic and final models shows the variation of the models (Figure 11).

### 3.2. Numerical Simulation

Aiming at defining the material model parameters suitable for the finite element analyses of FRC structures, the numerical simulation of tested beams was performed using GiD v12 and Atena Studio v5 software. Atena is an extremely powerful tool for non-linear analysis of reinforced concrete structures, allowing the representation of the real structural behavior of cementitious materials, while GiD software was used in this study as a pre-processor.

The numerical analyses were performed on 2D models representing the notched beams tested in laboratory investigation. Since the simulated problem was symmetrical, a half of the beam was considered in all analyses. The beams were modelled with the plane stress elements with a thickness of 0.15 midealization of plane stress. Notably, each beam was modelled in two separate parts with fixed connection between them: the unnotched part of the beam as well as the part of the beam above the initial notch (the dimensions of the notch: 2.5 × 25 mm^2^). The model division was crucial for obtaining the structured mesh. In order to avoid the stress concentration, the boundary conditions were transferred to the model through additional steel solid elastic plates with the dimensions of 10 × 10 mm^2^ (in the case of a support) or 2.5 × 10 mm^2^ (in the case of a loading). The bottom support of the beam was modelled as a constraint for point along the Y axis. Additionally, the line support with constrained displacement along the X axis was defined to represent the symmetry of the beams. The load was set as an increasing displacement along Y axis with 50 intervals of 0.1 mm divided into 500 loading steps. Moreover, two monitoring points were introduced into the model to observe the increasing over time of CMOD width as well as support reaction. The geometry and boundary conditions of the model are presented in Figure 12a. The mesh was regular and structured with elements size of 5 mm (Figure 12b). The isoparametric plane quadrilateral elements, with 4 or 9 Gauss integration points s for the case of bilinear or bi-quadratic interpolation, were used. Furthermore, the sensitivity analysis was performed on the numerical model with 1.25% of fibers by comparing the CMOD-load curves obtained for various sizes of mesh elements (Figure 13). With the sufficient mesh density, i.e., the element size of 20 mm, the size of mesh elements no longer affected the obtained results. This observation, combined with the fact that the use of mesh elements smaller than 5 mm significantly extended the calculation time, prompted us to apply 5 mm mesh for further analyses.

Atena software offers a rich library of material models available for concrete analysis, including a material model called CC3DNonLinCementitious2SHCC, used in this study for modelling of FRC composites. Roughly speaking, CC3NonLinCementitious2 models are fracture-plastic constitutive models combining fracturing behavior under tension and plastic behavior under compression. The hardening/softening plasticity model is associated with Menétrey-Willam failure surfaces, while the fracture model introduced Rankine failure criterion and exponential softening. The material model designed for FRC composites, i.e., CC3DNonLinCementitious2SHCC, involves modified tensile softening regime as well as shear retention factor [30]. The smeared cracking approach was applied in the investigation with the fixed crack model. In the model the crack is formed when the principal stress exceeds the tensile strength. It is also assumed that the cracks are uniformly distributed within the material volume. 

The initial simulation for each fiber content was performed using the material model parameters calculated according to MC2010 [13]. The results of the simulation were then precisely compared with the results from laboratory tests of FRC composites. Due to the unsatisfactory outcomes, the parameters of the material model in Atena were further modified and the inverse analysis was employed. Consequently, the modified stress-strain relationship for numerical simulations of FRC composites was proposed.

### 3.3. Results from Numerical Modelling

The results from numerical simulation are presented as the load-CMOD curves for each volume ratio of the fibers (Figure 14). The results are extensively compared with the load-CMOD relationships obtained in the laboratory investigation. It can be seen that the proposed modification of material model parameters led to high compliance between the experimental results and numerical simulation.

The crack location for beams containing 0.5% and 1.5% of RSF obtained for CMOD = 0.5 mm and CMOD = 2.5 mm are presented in Figure 15 and Figure 16, respectively. It can be clearly seen that the cracking pattern changes with the increase of the fiber volume ratio. 

Figure 15 and Figure 16 introduce the principal stress of the specimens reinforced with 0.5% and 1.5% of RSF for CMOD = 0.5 mm and CMOD = 2.5 mm, respectively. The stress distribution clearly indicates the overall response to the applied loads of the whole specimens with higher percentage of the fibers (see Figure 17 and Figure 18). Fundamentally, this phenomenon is attributed to the crack bridging effect of the fibers, affecting the other parts of the beam after cracking of the solid material. Indeed, such an effect is much more visible for the beams containing 1.5% of RSF.

## 4. Conclusions

This research provides the recommendations for the material model parameters essential to perform the numerical simulations of SRFRC. 

We first conducted the laboratory experiments by means of the three-point bending tests with different content per volume of recycled steel fibers in the range of 0.5–1.5%. Within the laboratory investigation, it was proved that the recycled steel fibers (steel cord) hold a great potential to enhance the residual flexural tensile strengths of concrete. The flexural mechanical parameters of concrete increased with the increase of the fiber volume ratio. However, the effectiveness of the fibers loadings below 1% was quite poor, what was also proved in the previous works. Nevertheless, the noticeable increase was observed for the highest content of fibers, i.e., 1.25% and 1.5%. As expected, the RSF do not significantly affect the compressive strength of concrete, which was equal to 40 MPa. 

We then performed the numerical simulation using GiD v12 and Atena Studio v5 software with the material model parameters proposed by MC2010 for FRC. The parameters of the MC2010 material model covers residual flexural tensile stresses and corresponding strains taken from the flexural tests of FRC. The unsatisfactory outcomes prompted us to modify the material model using the inverse analysis to meet the results of the laboratory investigation with the stress-strain curves obtained from the finite element method (FEM) analysis. A key modification of the material model was to replace the values of maximum tensile strength with the limit of proportionality. A very good compliance between the experimental results and numerical simulation of load-CMOD curves was obtained. 

The analyses have proved that the numerical simulations of SRFRC may be performed using the modified MC2020 material model, thus allowing a viable approach towards environmentally friendly building structures.

## Figures and Tables

**Figure 1 materials-14-01828-f001:**
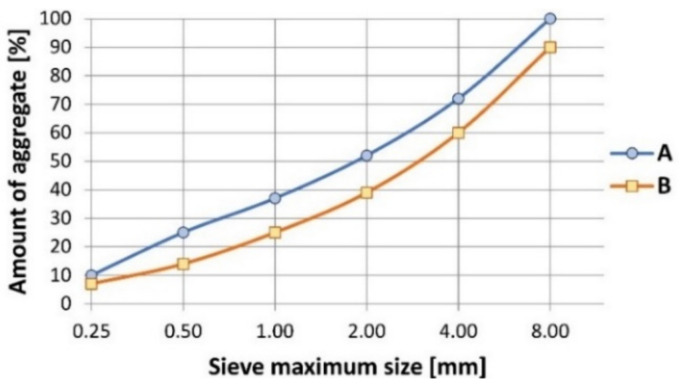
The aggregate fraction analysis.

**Figure 2 materials-14-01828-f002:**
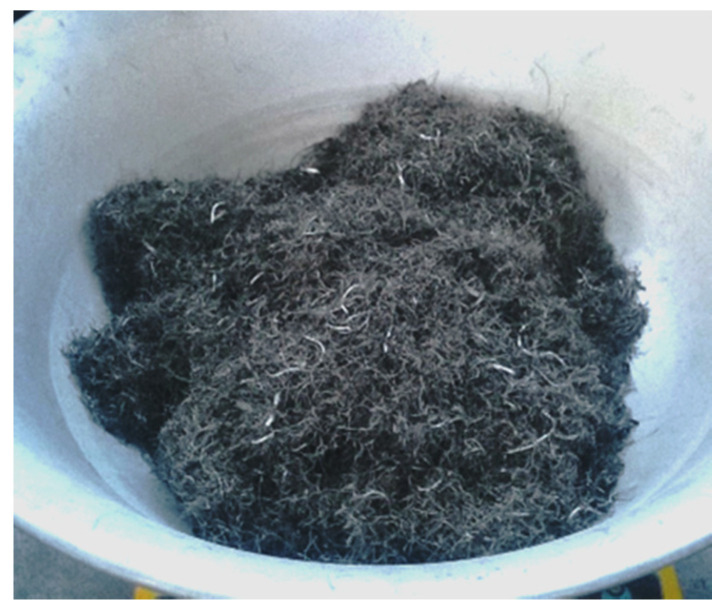
Recycled steel fibers used in the investigation.

**Figure 3 materials-14-01828-f003:**
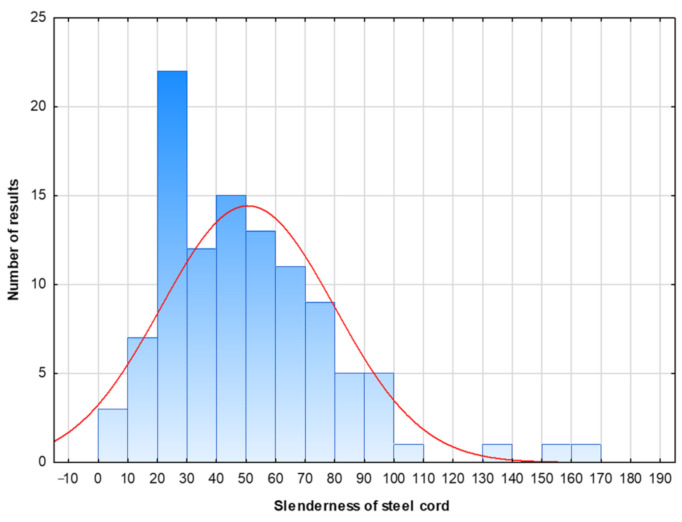
The slenderness of the steel cord.

**Figure 4 materials-14-01828-f004:**
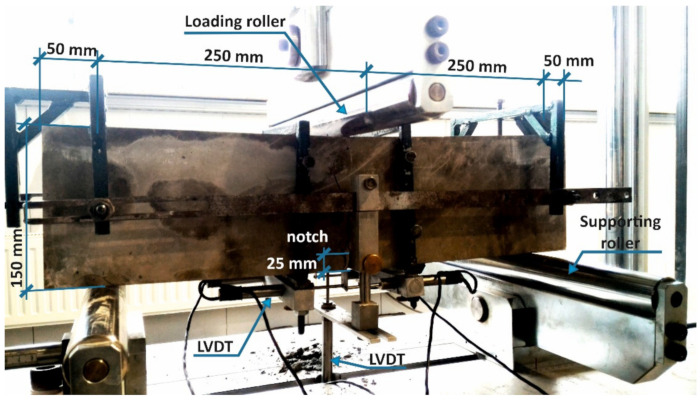
Geometry and static scheme of tested beams with the displacement sensors system for measuring the deflection and crack width (CMOD).

**Figure 5 materials-14-01828-f005:**
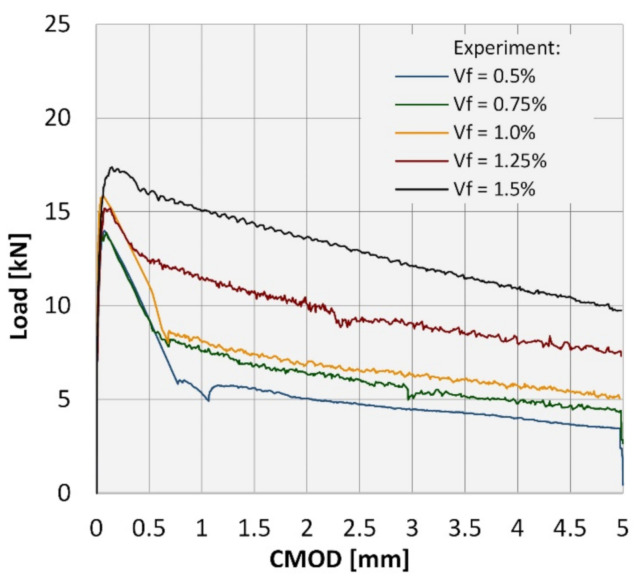
Average load-CMOD curves obtained in the flexural tests.

**Figure 6 materials-14-01828-f006:**
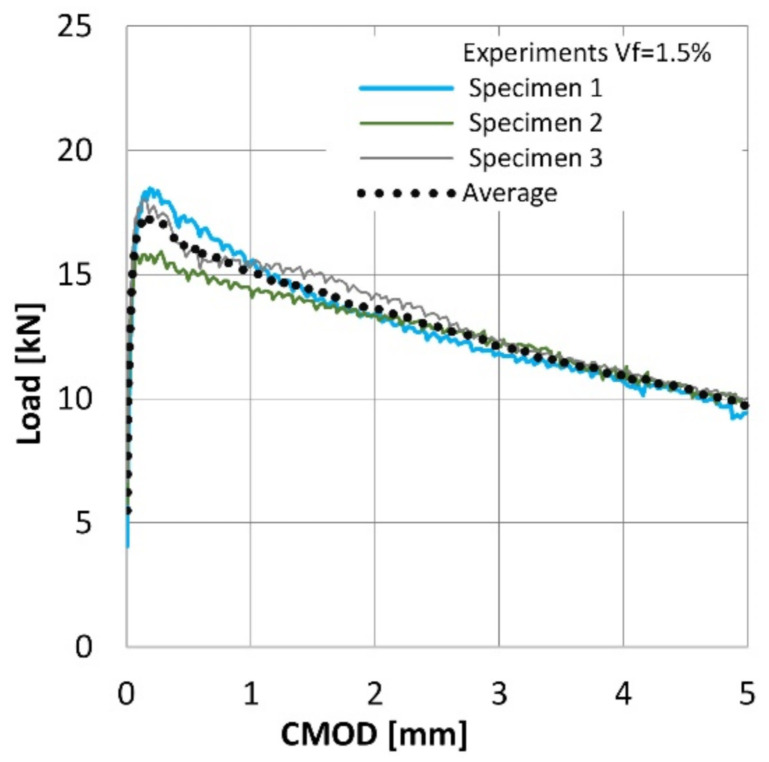
Load-CMOD curves for three specimens reinforced with 1.5% of recycled steel fibers (RSF).

**Figure 7 materials-14-01828-f007:**
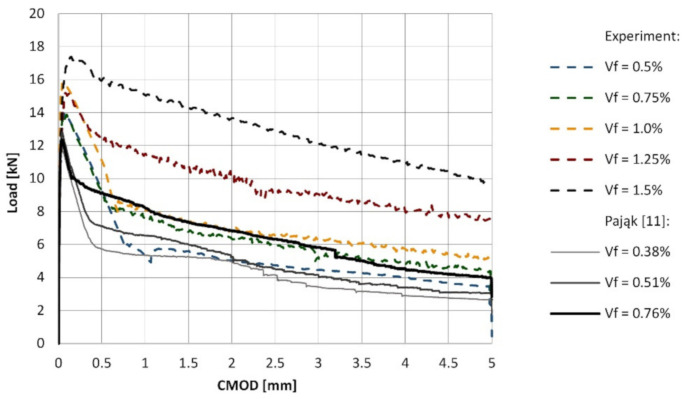
Average load-CMOD curves obtained in the flexural tests.

**Figure 8 materials-14-01828-f008:**
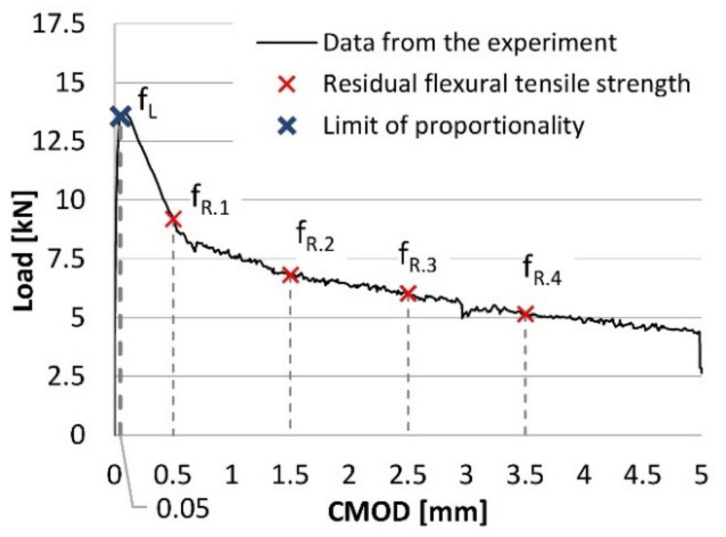
Determination of the flexural strengths from the load-CMOD relationship.

**Figure 9 materials-14-01828-f009:**
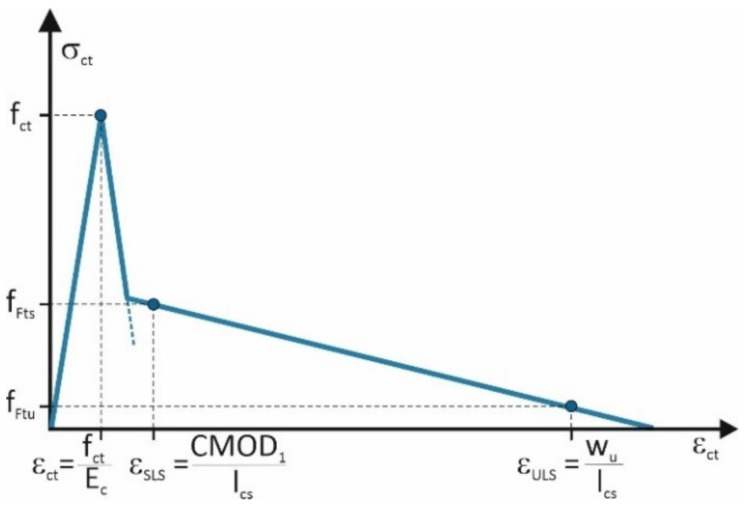
Material model for fiber reinforced concrete (FRC) according to MC2010.

**Figure 10 materials-14-01828-f010:**
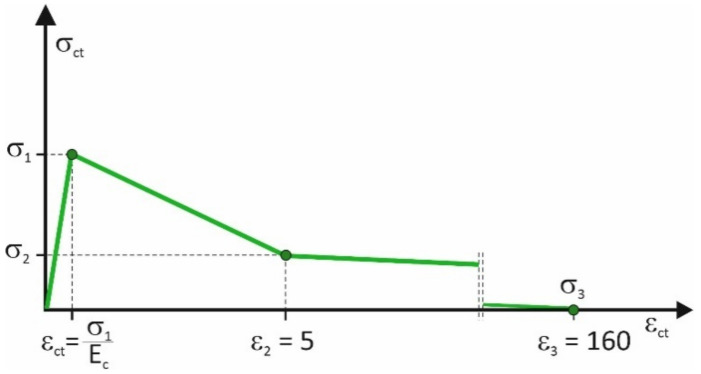
Assumptions of the final material model used in the numerical simulation.

**Figure 11 materials-14-01828-f011:**
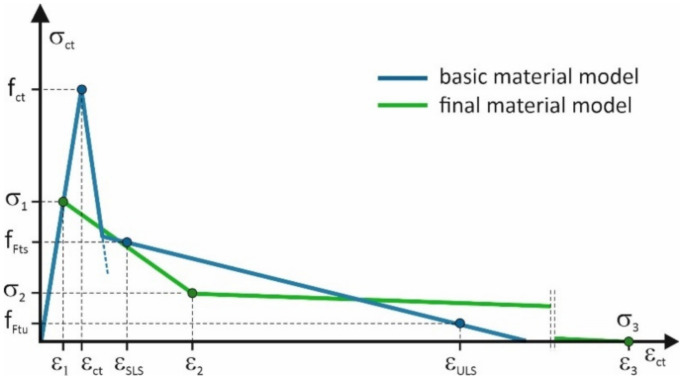
The comparison of the basic and final material models.

**Figure 12 materials-14-01828-f012:**
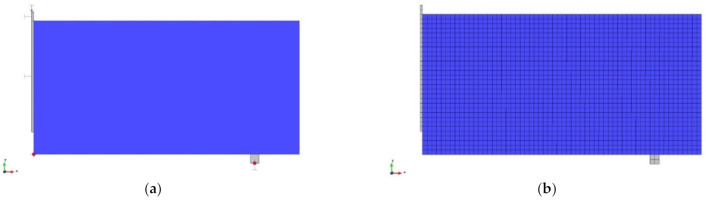
Numerical model of concrete beams: (**a**) the geometry and boundary conditions, (**b**) the model mesh.

**Figure 13 materials-14-01828-f013:**
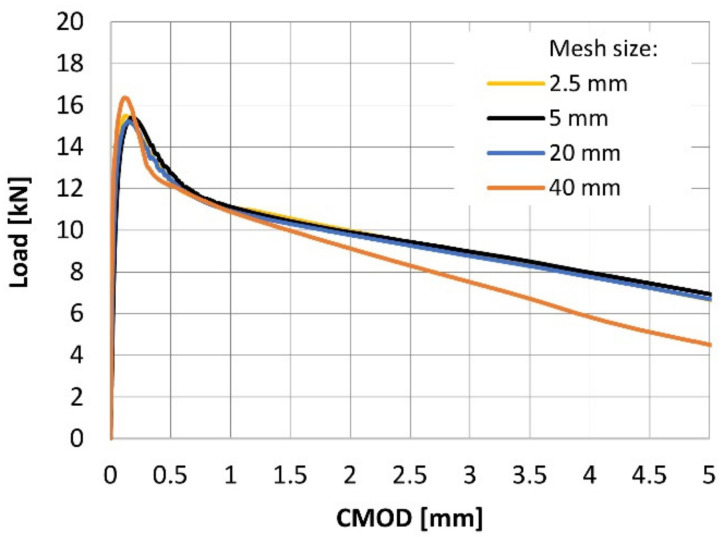
Mesh sensitivity study.

**Figure 14 materials-14-01828-f014:**
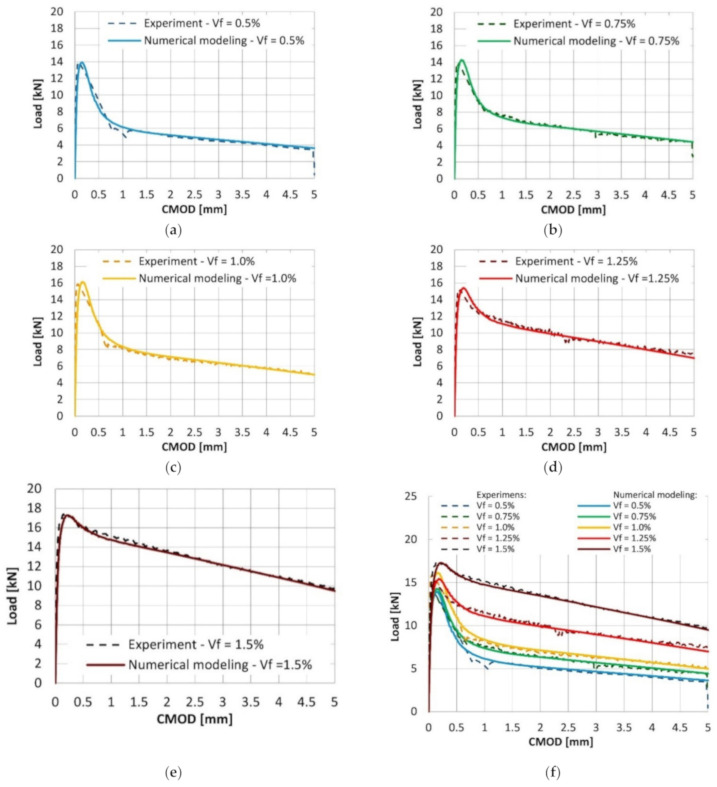
Results from the numerical simulation in comparison to the laboratory investigation of beams reinforced with fiber volume ratio equal to (**a**) 0.5%; (**b**) 0.75%; (**c**) 1.0%; (**d**) 1.25%; and (**e**) 1.5%. (**f**) Collation of the results.

**Figure 15 materials-14-01828-f015:**
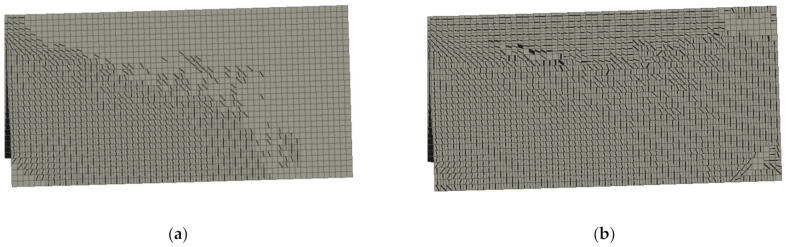
Crack propagation in all models for CMOD = 0.5 mm: (**a**) 0.5% and (**b**) 1.5%.

**Figure 16 materials-14-01828-f016:**
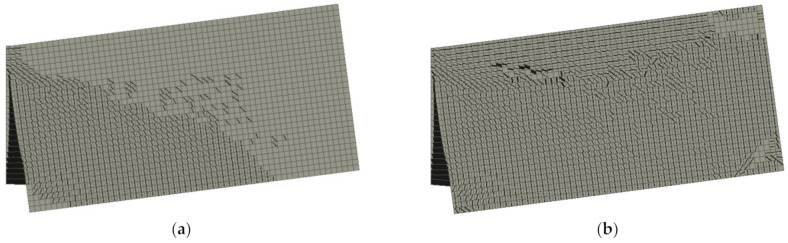
Crack propagation in all models for CMOD = 2.5 mm: (**a**) 0.5% and (**b**) 1.5%.

**Figure 17 materials-14-01828-f017:**
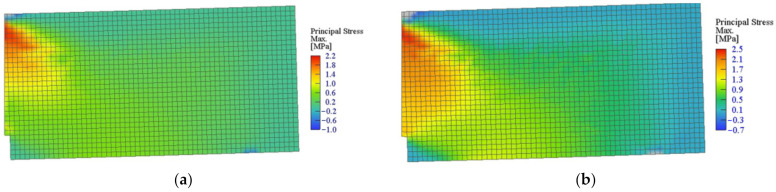
Maximum principal stress in all models for CMOD = 0.5 mm: (**a**) 0.5% and (**b**) 1.5%.

**Figure 18 materials-14-01828-f018:**
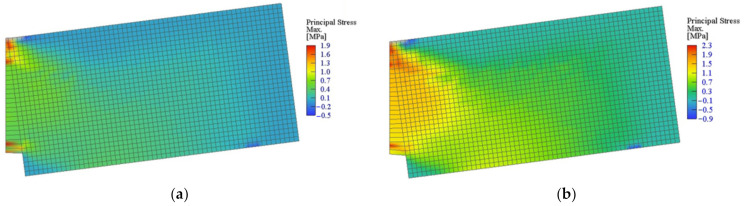
Maximum principal stress in all models for CMOD = 2.5 mm: (**a**) 0.5% and (**b**) 1.5%.

**Table 1 materials-14-01828-t001:** Mix composition.

Cement CEM I 42.5 R (kg/m^3^)	Water (kg/m^3^)	Aggregate (kg/m^3^)	Fiber Reinforcement Ratio	w/c
350	192.5	1840	0.5%; 0.75%; 1.0%; 1.25%; 1.5%	0.55

**Table 2 materials-14-01828-t002:** The parameters of recycled steel fibers.

Parameter	Average (mm)	Standard Deviation (mm)	Median (mm)	Minimum Value (mm)	Maximum Value (mm)
Diameter	0.306	0.183	0.250	0.050	1.350
Length	13.29	5.80	12.57	2.90	27.90

**Table 3 materials-14-01828-t003:** Results from fresh concrete tests.

Mix	S (mm)	Degree of Compactability *c*	Fresh Concrete Consistency Class
0.5%	25	1.07	C3
0.75%	20	1.05	C3
1.0%	35	1.10	C3
1.25%	45	1.13	C2
1.5%	45	1.13	C2

**Table 4 materials-14-01828-t004:** Results of the flexural tests.

Mix	Compressive Strength (Cylinders) (MPa)	Limit of Proportionality (MPa)	Residual Flexural Tensile Strength (MPa)				f_R.3_/f_R.1_	f_R.1_/f_LOP_
		f_LOP_	f_R.1_	f_R.2_	f_R.3_	f_R.4_		
0.5%	39.88	4.31	2.97	1.79	1.52	1.36	0.51	0.69
0.75%	40.29	4.33	2.94	2.17	1.93	1.65	0.66	0.68
1.0%	41.48	5.03	3.55	2.37	2.09	1.94	0.59	0.71
1.25%	40.08	4.47	3.97	3.43	3.03	2.72	0.76	0.89
1.5%	42.79	4.85	5.13	4.60	4.13	3.67	0.81	1.06

**Table 5 materials-14-01828-t005:** The parameters of the basic material model [20] with the data from the laboratory investigation.

Mix	Data from Laboratory Investigation			Basic Material Model					
	Compressive Strength f_cm_ (MPa)	Residual Flexural Tensile Strength (MPa)		Tensile Strength (MPa)	Flexural Tensile Strength (MPa)		Tensile Strain		
		f_R.1_	f_R.3_	f_ctm_	f_Fts_	f_ftu_	ε_ct_	ε_SLS_	ε_ULS_
0.5%	39.88	2.97	1.52	3.50	1.34	0.16	0.0719	4	20
0.75%	40.29	2.94	1.93	3.53	1.32	0.38	0.0721	4	20
1.0%	41.48	3.55	2.09	3.59	1.60	0.34	0.0728	4	20
1.25%	40.08	3.97	3.03	3.51	1.79	0.72	0.0720	4	20
1.5%	42.79	5.13	4.13	3.67	2.31	1.04	0.0736	4	20

**Table 6 materials-14-01828-t006:** The parameters of the final material model with the data from the laboratory investigation.

Mix	Data for the Final Material Model		Final Material Model					
	Limit of Proportionality f_L_ (MPa)	Residual Flexural Tensile Strength for CMOD = 2.5 mm (MPa) f_R.3_	Tensile Stress (MPa)			Tensile Strain		
			σ_1_	σ_2_	σ_3_	ε_1_	ε_2_	ε_3_
0.5%	4.28	1.52	1.96	0.58	0	0.0603	5	160
0.75%	4.33	1.93	1.98	0.72	0	0.0608	5	160
1.0%	5.04	2.09	2.30	0.81	0	0.0701	5	160
1.25%	4.47	3.03	2.04	1.16	0	0.0628	5	160
1.5%	4.86	4.13	2.22	1.58	0	0.0668	5	160

## Data Availability

The data presented in this study are available on request from the corresponding author.

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
