# Peer review of "Laboratory Investigation and Numerical Modelling of Concrete Reinforced with Recycled Steel Fibers"

_materials, 2021, doi:10.3390/ma14081828_

Round 1
Reviewer 1 Report
The paper is quite interesting, and it deserves to be published on Materials.
Minor recommendations are reported:
1)I suggest reporting the paper's outline at the end of the instruction section, discussing the gap that you want to fill.
2)The conclusion section can be improved with more considerations.
3)I suggest referencing more paper about concrete structures, e.i:
10.1016/j.ijheatmasstransfer.2016.03.033
10.1016/S0013-7944(02)00153-4
10.3390/FIB8060042
4) The numerical part needs to be improved; for instance, what kind of finite element did you use? Shape function?
5) A sensitivity analysis can be useful to understand the effectiveness of the numerical procedure.
Author Response
Response to Reviewer 1 Comments
We are grateful the Reviewer for the comments. We have answered each of the points below:
Point 1: I suggest reporting the paper's outline at the end of the instruction section, discussing the gap that you want to fill.
Response 1: The comment was added to the introduction:
“The numerical simulations of FRC are based mainly on sophisticated numerical tools. The aim of the present paper was to prepare the useful and relatively uncomplicated numerical tool for performing the analysis of concrete reinforced with RSF. The Atena Studio v5 appeared to be the best program for this purpose. By modification of the material model based on laboratory investigation the RSFRC structures can be easily analyzed. Thus, just knowing the residual flexural tensile strengths from the laboratory investigation the RSFRC structures can be analyzed. Noteworthy, this approach may open up new avenues for the technological applications of RSFRC in construction.”
Generally, the large part of the introduction was revised.
Point 2: The conclusion section can be improved with more considerations.
Response 2: The conclusions were extended.
Point 3: I suggest referencing more paper about concrete structures, e.i:
10.1016/j.ijheatmasstransfer.2016.03.033
10.1016/S0013-7944(02)00153-4
10.3390/FIB8060042
Response 3: The references were cited in the paper in the Introduction. Thank you for pointing the valuable works in the topic.
Point 4: The numerical part needs to be improved; for instance, what kind of finite element did you use? Shape function?
Response 4: The comment was added to the text:
“Element type used was plane quadrilateral elements. These are isoparametric elements integrated by Gauss integration at 4 or 9 points for the case of bilinear or bi-quadratic interpolation.”
Point 5: A sensitivity analysis can be useful to understand the effectiveness of the numerical procedure.
Response 5: The comment and the figure was added to the text:
“The isoparametric plane quadrilateral elements with 4 or 9 Gauss integration points s for the case of bilinear or bi-quadratic interpolation, were used. Furthermore, the sensitivity analysis was performed on the numerical model with 1.25% of fibres by comparing the CMOD-load curves obtained for various sizes of mesh elements. With the sufficient mesh density, i.e. the element size of 20 mm, the size of mesh elements no longer affects the obtained results. This observation, combined with the fact that the use of mesh elements smaller than 5 mm significantly extends the calculation time, prompted us to apply 5 mm mesh for further analyses.”

Reviewer 2 Report
Paper title:
Laboratory investigation and numerical modelling of concrete reinforced with recycled steel fibers
Authors: PajÄ…k, M. Krystek, M. Zakrzewski, J. Domski
The paper is interesting. The manuscript is well written.
The use of steel fiber reinforced concrete is a well-known tool for the enhancement of the capacities of concrete under tension. The use of recycled steel fibers produced from car tires is a new very interesting approach towards the environmental-friendly construction.
Experimental research is the main research tool in this direction. Flexural tensile strength tests are presented according to EN-14651; 15 samples were prepared and tested.
Further, the main goal of the submitted work is the numerical simulation of recycled steel fibers reinforced concrete elements.
Obviously, due to the fact that experimental procedures are expensive, developing finite element modeling procedures regarding the study of RC members has become absolutely important. Therefore, the attempted finite element approach may be proved useful for the practicing engineer as well as for the researchers.
However, comments and conclusions of the submitted paper have to be elaborated before publication.
- The literature background reported in Introduction is rather informative. However, the state-of-the-art related to the flexural and cracking response and the analysis of beams with steel-fibers is not adequately highlighted. The manuscript partially fails to build on the existing works and to establish the relevance of the work reported in the paper. Therefore, additional quite relative studies are suggested to be considered and commented properly in order to promote the findings and the objectives of this paper.
- Research significance, novelty and subsequent impact of the study on the state of the practice should be further highlighted. An improvement of the systematic literature review provided in introduction would help in this direction.
- The application of tensile models behaviour for steel-fiber reinforced concrete with tension softening part is a very important feature of the submitted work. Stress-strain curves are presented in figures 7 and 9. Nevertheless, it has to be considered that the influence of fibers in the FEM analyses is not really explained in the manuscript. A good attempt is presented in paragraph 3.1.
-About the use of finite elements and the sensitivity analysis. The mesh is presented in figure 11; but more details are needed. A sensitivity analysis of the used finite element mesh has to be added in the manuscript. The number of the applied finite elements versus the accuracy of the results has to be studied. Comments about the relationship between the mesh density and the yielded results are definitely welcome. These comments would really increase the validity of the presented approach.
- The tension softening concept used in the study for the simulation of the cracking of the specimens suggests that the finite element smeared cracking approach is applied (or otherwise what?). Some comments are definitely required about the applied smeared cracking approach.
- In any case more comments about the used finite elements (a description) have to be included in the revised manuscript. What about the cracking (smeared cracking?).
- The conclusions are qualitatively rather correct.
- The fiber amounts are per volume (volume ratio); it has to be notified in abstract (and conclusions).
- In general, the reference list can be considered as rather poor. Additional strongly suggested references based on the aforementioned comments:
“Effect of Steel Fibers on the Hysteretic Performance of Concrete Beams with Steel Reinforcement—Tests and Analysis” materials, 2020, Vol. 13, (2923).
“Analytical approach for the evaluation of minimum fiber factor required for steel fibrous concrete beams under combined shear and flexure”. Construction and Building Materials, 2013, 43, 317–336.
- A notation list (including the abbreviations) is needed.
Final Conclusion
The submitted paper is a useful work that has to be accepted after a careful revision.
Author Response
Response to Reviewer 2 Comments
We are grateful the reviewer for the comments. We have answered each of the points below:
Point 1: The literature background reported in Introduction is rather informative. However, the state-of-the-art related to the flexural and cracking response and the analysis of beams with steel-fibers is not adequately highlighted. The manuscript partially fails to build on the existing works and to establish the relevance of the work reported in the paper. Therefore, additional quite relative studies are suggested to be considered and commented properly in order to promote the findings and the objectives of this paper.
Response 1: The literature background was added to the Introduction. The large part of the introduction was revised.
Point 2: Research significance, novelty and subsequent impact of the study on the state of the practice should be further highlighted. An improvement of the systematic literature review provided in introduction would help in this direction.
Response 2: The comment was added to the Introduction:
“The numerical simulations of FRC are based mainly on sophisticated numerical tools. The aim of the present paper was to prepare the useful and relatively uncomplicated numerical tool for performing the analysis of concrete reinforced with RSF. The Atena Studio v5 appeared to be the best program for this purpose. By modification of the material model based on laboratory investigation the RSFRC structures can be easily analyzed. Thus, just knowing the residual flexural tensile strengths from the laboratory investigation the RSFRC structures can be analyzed. Noteworthy, this approach may open up new avenues for the technological applications of RSFRC in construction.”
Point 3: The application of tensile models behaviour for steel-fiber reinforced concrete with tension softening part is a very important feature of the submitted work. Stress-strain curves are presented in figures 7 and 9. Nevertheless, it has to be considered that the influence of fibers in the FEM analyses is not really explained in the manuscript . A good attempt is presented in paragraph 3.1.
Response 3: Strain Hardening Cementitious Composite material is suitable for fiber reinforced concrete. This model enables to entry the exact values of the characteristic points of the softening branch of the load-CMOD curves. The data which were taken from the experiments are tensile stresses and tensile strains (Fig. 9). In the table 6 the parameters are summarized.
Point 4: About the use of finite elements and the sensitivity analysis. The mesh is presented in figure 11; but more details are needed. A sensitivity analysis of the used finite element mesh has to be added in the manuscript. The number of the applied finite elements versus the accuracy of the results has to be studied. Comments about the relationship between the mesh density and the yielded results are definitely welcome. These comments would really increase the validity of the presented approach.
Response 4:The comment and the figure were added to the text:
“The isoparametric plane quadrilateral elements with 4 or 9 Gauss integration points s for the case of bilinear or bi-quadratic interpolation, were used. Furthermore, the sensitivity analysis was performed on the numerical model with 1.25% of fibres by comparing the CMOD-load curves obtained for various sizes of mesh elements. With the sufficient mesh density, i.e. the element size of 20 mm, the size of mesh elements no longer affects the obtained results. This observation, combined with the fact that the use of mesh elements smaller than 5 mm significantly extends the calculation time, prompted us to apply 5 mm mesh for further analyses.”
Point 5: The tension softening concept used in the study for the simulation of the cracking of the specimens suggests that the finite element smeared cracking approach is applied (or otherwise what?). Some comments are definitely required about the applied smeared cracking approach.
Response 5: The Reviewer is of course right. The smeared cracking approach was applied in the investigation with the fixed crack model. The suitable comment was put in the text.
Point 6: In any case more comments about the used finite elements (a description) have to be included in the revised manuscript. What about the cracking (smeared cracking?).
Response 6: The comment was added to the text:
“The smeared cracking approach was applied in the investigation with the fixed crack model. In the model the crack is formed when the principal stress exceeds the tensile strength. It is also assumed that the cracks are uniformly distributed within the material volume.”
Point 7: The fiber amounts are per volume (volume ratio); it has to be notified in abstract (and conclusions).
Response 7: It was revised.
Point 8: In general, the reference list can be considered as rather poor. Additional strongly suggested references based on the aforementioned comments:
Response 8: “Effect of Steel Fibers on the Hysteretic Performance of Concrete Beams with Steel Reinforcement—Tests and Analysis” Materials, 2020, Vol. 13, (2923).
“Analytical approach for the evaluation of minimum fiber factor required for steel fibrous concrete beams under combined shear and flexure”. Construction and Building Materials, 2013, 43, 317–336.
The references were added to the paper. Thank you for pointing the valuable works in the topic.
Point 9: A notation list (including the abbreviations) is needed.
Response 9: The paper contains the list of references.

Reviewer 3 Report
Dear authors,
This study shows laboratory results and numerical analysis of concrete reinforced with waste steel fibres obtained during the process of recycling discarded tires.
I appreciate the practical contribution of the article, a detailed description of the regressive analysis of the data obtained from the tests, as well as the graphic design of all outputs.
I understand the use of "waste sand" as an aggregate. What properties of concrete do you expect (from the point of view of your tests), if only a standard aggregate would be used?
Will you test similar concretes in the future?
Did I understand well that the fine aggregate fraction is 0-4 and the coarse aggregate fraction is 0 to 8?
What type the coarse aggregate is?
Your work (PajÄ…k, Matec, 2019) contains a mixture of sand and coarse aggregate (size up to 16) - even here I see a difference from the presented work and I am interested in the reason. (I mean the information from lines 142 to 145, where you point out the differences - are they not caused by a different aggregate and its size?)
Do you have any background (literature) where it is proven that the use of recycled steel fibres from tires is cheaper than standard steel fibres (I mean also in terms of the ratio of production cost to mechanical properties)?
A few minor language adjustments could be made - for example:
line 40 - the replacement
line 51 - the steel
line 80 - has
line 93 - the steel
line 116 - a variable
line 215 - the compliance of the most precise results
line 220 - breakpoint
and several others throughout the article.
A small note at the end:
you use the words "behavior" and "fibers" - a non-British variant - and you use the words "modelling" and "modelled" - a non-American variant. This may be the standard in your country, but it should probably be uniform.
Best regards
Author Response
Response to Reviewer 3 Comments
Dear Reviewer,
Thank you for your comments. We have answered each of the points below:
Point 1: I understand the use of "waste sand" as an aggregate. What properties of concrete do you expect (from the point of view of your tests), if only a standard aggregate would be used?
Will you test similar concretes in the future? Did I understand well that the fine aggregate fraction is 0-4 and the coarse aggregate fraction is 0 to 8? What type the coarse aggregate is?
Response 1: If only a standard aggregate would be used we expect that properties of concrete will be similar as tested concrete. The name “waste sand” is due to the fact that in the region where we live we have very large deposits of sand and very small deposits of coarse aggregate. Waste sand is formed during the process of hydroclassifying the material in natural aggregate mines. During this process, coarse aggregate is extracted and sand is waste. In this process, very large amounts of sand are obtained. On the other hand, transporting waste sand over longer distances is unprofitable.
The fine aggregate fraction was 0-2. This is due to the fact that such fractions of waste sand are extracted in the largest amounts. The coarse aggregate fraction is 2 to 8. The coarse aggregate are natural stones.
Point 2: Your work (PajÄ…k, Matec, 2019) contains a mixture of sand and coarse aggregate (size up to 16) - even here I see a difference from the presented work and I am interested in the reason. (I mean the information from lines 142 to 145, where you point out the differences - are they not caused by a different aggregate and its size?)
Response 2: Thank you for this observation. The comment presented in the paper in the previous lines 142-145 concerned only the scatters between the results. It was probably caused by the size and type of the aggregate and by the different slump flow of these two concretes.
Further, we didn’t took too much attention to comparison of these too works in the paper. Maybe it is worth that because we have obtained a very good compliance of the results. Please note, that the laboratory investigations were performed in two different laboratories (Gliwice, Koszalin). The paper was enriched with the comparison of the results from these two works considering the proceeding of the load-deflection curves. It could be clearly seen that the response of RSFRC after the CMOD around 0.5 mm was comparable, even though the concrete with the higher compressive strength (around 50MPa) was tested in previous work. The comment and the figure with the comparison of the results from both of these tests were added to the paper:
“This fact may be attributed to the size and type of the aggregate and the slump flow of the mix in case of the concrete described in the work [11]. The size of the coarse aggregate (16 mm) and the slump flow was bigger than in the case of the results presented herein. Generally, the proceeding of the load-CMOD curves noted in the case of those two laboratory investigation were comparable for the related fibre volume ratios, even though the concrete with the higher compressive strength (around 50MPa) was tested in [11].”
Point 3: Do you have any background (literature) where it is proven that the use of recycled steel fibres from tires is cheaper than standard steel fibres (I mean also in terms of the ratio of production cost to mechanical properties)?
Response 3: As far as the authors are concerned there is no paper facing the problem of costs. It may be attributed to the completely different technologies of possessing the fibres in different countries what affects their prise. Every company protect their know-how and the real costs of recovering fires from tires.
Point 4: A few minor language adjustments could be made - for example:
line 40 - the replacement
line 51 - the steel
line 80 - has
line 93 - the steel
line 116 - a variable
line 215 - the compliance of the most precise results
line 220 - breakpoint
and several others throughout the article.
A small note at the end:
you use the words "behavior" and "fibers" - a non-British variant - and you use the words "modelling" and "modelled" - a non-American variant. This may be the standard in your country, but it should probably be uniform.
Response 4: The language adjustement has been done.
The „fibers” were change to „fibres”.
The „behavior” were change to „behaviour”.

Round 2
Reviewer 1 Report
The paper can be accepted in the present form